# Appropriate Lifelong Circadian Rhythms Are Established During Infancy: A Narrative Review

**DOI:** 10.3390/clockssleep7030041

**Published:** 2025-08-07

**Authors:** Teruhisa Miike

**Affiliations:** School of Medicine, Kumamoto University, Kumamoto 860-8556, Japan; t.miike4297@gmail.com

**Keywords:** circadian rhythm consolidation, sleep/wake rhythm, pregnant mother, fetus, infant, suprachiasmatic nucleus, DOHaD

## Abstract

In humans, the master circadian clock, present in the suprachiasmatic nucleus, plays an important role in controlling life-sustaining functions. The development of the circadian clock begins in the fetal period and is almost completed during infancy to early childhood, based on the developmental program that is influenced by the mother’s daily rhythms and, after birth, with the addition of information from the daily life environment. It is known that circadian rhythms are deeply related not only to the balance of a child’s mental and physical development but also to maintaining mental and physical health throughout one’s life. However, it has been suggested that various health problems in the future at any age may be caused by the occurrence of circadian disturbances transmitted by the mother during the fetal period. This phenomenon can be said to support the so-called DOHaD theory, and the involvement of the mother in the maturation of appropriate and stable circadian rhythms cannot be ignored. We consider the problems and countermeasures during the fetal and infant periods, which are important for the formation of circadian clocks.

## 1. Introduction

The basic functions related to the maintenance of daily human life are controlled by endogenous rhythms with a cycle of approximately 24 h, which are called circadian rhythms. Humans become sleepy, wake up, feel hungry, and eat at roughly the same time every day. This natural rhythm and schedule of life that everyone follows is due to the circadian clocks that are present in the human body [1,2]. As the circadian rhythm is observed even in constant environmental conditions when there is no change in light or temperature, it is clear that living organisms have a clock mechanism in their bodies called the “biological clock.” The circadian rhythm is an oscillation with a cycle of approximately 24 h, which has been observed to participate in almost all physiological functions of the human brain and body [1]. In particular, the central mechanism of the human circadian rhythm biological clock, present in the suprachiasmatic nucleus (SCN), controls life-sustaining rhythmic functions such as the daily rhythms of the sleep/wake cycle [3,4,5], thermoregulation [6,7], hormone secretion [8], energy metabolism [8,9,10,11,12,13], food intake [14,15,16], immune function [17,18], autonomic nervous function [19,20], cerebellum and motor coordination [21], gut microbiota [22], and brain function [23,24,25,26] (Figure 1). The circadian clock controls many important aspects of physiology, from sleep–wake cycles to metabolism [26] (Figure 1). In addition to circadian rhythm, biological clocks include rhythms with cycles shorter than 24 h (ultradian rhythm, UR), rhythms of approximately half a day (12 h), such as those observed during naps (circasemidian rhythms), circatidal rhythms (12.4 h), rhythms with cycles of approximately one month (circalunar rhythms = approximately 1 month), and circannual rhythms (approximately one year). For more details on these biological rhythms, please refer to the many well-known specialist books published to date on these topics [27,28,29,30].

This review focuses only on the relationship between the circadian rhythm, which is the most relevant rhythm to human life, and the ultradian rhythm, which harbors/hosts/contains the master clock and is present by the middle of pregnancy in human and non-human primate fetuses [31,32].

After birth, as the fetus exposure to light and dark, the formation of the central clock in the SCN, begins, until almost complete maturation by early infancy (1–2 years of age) [3,33,34].

The best-known biological clock is the circadian clock, which actively changes the internal environment to match the approximately 24-h light–dark cycle caused by the rotation of the Earth. The human circadian clock was once thought to be about 25 h; however, in 1999, it was reported that the biological human circadian rhythm is 24.18 h (being shorter in some people) [35].

Fortunately, humans have a system (entrainment mechanism) that synchronizes the timing of their biological clocks to the external 24-h light–dark cycle, although there is a slight deviation associated with the duration of Earth’s rotation [36].

Mammalian circadian rhythms are controlled by endogenous biological oscillators, including a master clock located in the hypothalamic SCN. The duration of this oscillation is approximately 24 h, so to synchronize the circadian rhythm with the environment, it must be entrained daily by Zeitgeber (time provider) signals such as light–dark cycles, food, temperature, and social interactions [37]. The important thing is that we live on Earth in modern society, interacting with schools and society.

## 2. Two Circadian Clocks

There are two main types of circadian clocks: (1) the central clock (also called the master clock) [1,38,39,40] and (2) the peripheral clock [41,42]. As almost all cells in the human body have clock-like functions [43,44], some people consider the intracellular clock to be the third clock. However, in general, the intracellular clock is often interpreted as being included in the peripheral clock. In mammals, the 24-h rhythm of physiological functions and behavior is controlled by the master clock located in the SCN and is synchronized by a “slave” oscillator of similar molecular composition located in most cells in the body [45]. In recent years, increasing evidence has shown that the peripheral clock is also controlled by light and hormones independently of the SCN, and it has been reported that the peripheral clock has its own clock system and generates metabolic and physiological rhythms [46]. In any case, these two major clocks communicate with each other and cooperate to control the body’s rhythms [47]. Therefore, how these clocks work together is important for maintaining balanced function throughout the body.

## 3. The Process of Circadian Clock Development

### 3.1. Formation of Fetal Ultradian Rhythm

The development of the ultradian rhythm center in the pons and medulla oblongata begins around the 28th to 30th week of fetal development [48,49]. The circadian rhythm is based on an appropriate ultradian rhythm [50]. Among biological rhythms, rhythms with a period of several tens of minutes to several hours (up to 20 h) that is shorter than a 24-h period are called ultradian rhythms. They are also related to the development of sleep (mainly REM sleep) and may serve as the basis of the circadian rhythm formed after birth. In order to form healthy circadian clocks from fetal development to postnatal life, it is important that the ultradian rhythm is firmly formed around the 30th week of gestation and that the associated REM sleep develops appropriately, which leads to the smooth development of circadian rhythm after birth [48,50]. The ultradian rhythm control sites (the pons and medulla oblongata) begin to function around 28–30 weeks and mature around 37 weeks of pregnancy [27]. During fetal life, the cycle of REM sleep (active) and non-REM sleep (quiet) is basically a 40–50 min ultradian rhythm [49].

### 3.2. Formation of Circadian Rhythms During Fetal Development

In order to maintain brain function that can adapt to modern school/social life rhythms after birth, humans begin to adjust their daily rhythms to adapt to social life even during fetal development [51]. The formation of the circadian clock begins around the 20th to 22nd week of fetal development, when diurnal variations appear in heart rate, fetal movements, and respiratory movements, and circadian clocks appear in cells, tissues, and organs throughout the body [52]. However, the rhythms of each of these organs are not unified throughout the body, and each operates autonomously [52,53]. As the fetus is a part of the mother’s body and grows according to the mother’s daily rhythm, the mother’s lifestyle naturally affects the formation of the fetus’s own biological clock during pregnancy [51]. The intrauterine environment is rhythmic in nature, and the fetus is affected by changes in temperature, substrate, and the circadian rhythms of various maternal hormones. Meanwhile, the fetus develops an endogenous circadian system to prepare for life in an external environment where light, food availability, and other environmental factors change repeatedly and predictably every 24 h [54].

### 3.3. From Ultradian to Circadian Rhythms

Newborn babies up to one month of age sleep for about 14–17 h a day [55], with almost no distinction between day and night. During the newborn period, the circadian rhythm of sleep and wakefulness has not yet been fully formed, such that the daily life of the newborn follows an ultradian rhythm, a sleep/wake rhythm with a cycle of approximately 3–4 h [48,56,57,58]. This rhythm is important for newborns, and, as mentioned above, the formation of this rhythm becomes more stable, and the formation of the next circadian rhythm becomes smoother [48]. It is gradually becoming clear that the formation of the circadian clock is closely related to brain development, and it is known that many children with autism spectrum disorder (ASD) have poor formation of ultradian rhythms in the neonatal period [59].

However, the circadian clock that functions autonomously in each tissue and organ is not uniform in each individual and disappears at birth. Thus, the sleep/wake rhythm is not originally a circadian rhythm but, instead, an ultradian rhythm that is shorter than the circadian rhythm. The sleep/wake rhythm, which is only found in relatively advanced organisms with brains, is of recent origin and therefore does not belong to the original circadian rhythm. It takes several weeks after birth for the circadian rhythms and the sleep/wake rhythm to correlate, and until then, babies live with the original sleep/wake rhythm (ultradian rhythm). Shortly after birth, the baby begins to experience day and night (light and dark), and the formation of the central circadian clock in the SCN progresses. After the neonatal period, the circadian clock begins to develop rapidly. Eventually, the sleep/wake rhythm gradually comes under the control of the circadian rhythm throughout daily life and is incorporated into the 24-h circadian rhythm by 3–4 months of age. Early infancy is considered to be a critical period when all the organ, tissue, and cell clocks that were formed during fetal development and that operate autonomously are unified in the SCN and begin to function as a coordinated clock [33].

## 4. Factors Involved in the Formation of the Biological Clock

### 4.1. Fetal Period

#### 4.1.1. Living Environment of the Mother During Pregnancy

As the formation of the biological clock begins during fetal development, it is assumed that it is naturally influenced by the mother. In fact, the fetus—which is considered to be part of the mother’s body—is thought to live a life guided by the daily time-cues given by the mother, such as bedtime, wake-up time, and mealtime [33]. In other words, the fetus is considered to be in a similar state to the mother’s peripheral organs, such as the heart, liver, and limbs, and it is necessary to consider that the mother’s daily life is directly connected to the fetus’s daily life [51,60,61,62,63]. After birth, the fetus leaves the mother’s protection and transitions to life on Earth, so the uterus is provided with an appropriate environment for this transition. For example, the fetus’s internal rhythm is synchronized by various signals such as the mother’s body temperature, food, and melatonin (transferred from the mother to the fetus) [64,65,66].

Given this background, it has been reported that an inappropriate environment during pregnancy increases the risk of a child developing several chronic diseases in the future [66,67]. In other words, it has become clear that the predisposition to “future health problems” inherited from the mother is acquired during pregnancy. Therefore, it is thought that the most important thing for the child’s physical and mental development and future health maintenance is to protect the mother and fetus during pregnancy.

The intrauterine environment is rhythmic in nature, and the fetus is affected by changes in temperature, substrate, and the circadian rhythms of various maternal hormones [54]. Therefore, for the fetus to successfully transition to extrauterine life, the mother must provide various humoral/biophysical signals that are time-integrated precisely. In humans, there are many situations that can disrupt the circadian rhythms, including shift work, international travel, insomnia, and circadian rhythm disorders (e.g., advanced/delayed sleep phase disorders), and there is a growing consensus that this disruption of the circadian rhythm can have detrimental consequences for an individual’s health and well-being [54,66,67]. Epidemiological studies have reported that pregnant women engaged in chronic shift work have an increased risk of preterm birth, low birth weight, and miscarriage [68,69,70,71].

Abnormalities in sleep, feeding, and work schedules may disrupt maternal rhythms, such as melatonin fluctuations, resulting in the desynchronization of the maternal SCN and peripheral oscillators, leading to the deleterious effects of shift work on the fetus [72]. Pregnant rats exposed to simulated shift work (the complete reversal of the light–dark cycle every 3–4 days for several weeks) showed significantly reduced weight gain during early gestation, reduced fat pad and liver weights, and reduced amplitude of rhythms pertaining to corticosterone, glucose, insulin, and leptin levels [73]. Exposure of pregnant non-human primates to constant light suppresses the emergence of melatonin and body temperature rhythms in postnatal offspring [63,73,74,75,76,77,78]. This suggests that maternal rhythms are important for the early fetal brain development of circadian rhythms [63].

Changes in the circadian rhythm of gene expression are also associated with spatial memory impairments in these offspring, and these memory impairments can be prevented by the regular supplementation of melatonin to the mother [63].

In fact, disturbances such as nutritional problems during pregnancy, poor lifestyle habits, and exposure to infectious diseases and pollutants (including tobacco and alcohol) may affect the fetus. These influences from the mother are closely related to the child’s development, and pregnant mothers should avoid staying up late as much as possible, rest during the day, maintain a regular sleep/wake rhythm, and eat regular meals [14,15,16,42,79].

In particular, because the secretion of melatonin by the mother plays a crucial role in regulating these rhythms [63,74,75,76,77,78,79], maternal melatonin supplementation has been discussed, and its effectiveness has been reported [63,78,80,81]. Regular supplementation of the mother with melatonin can prevent these disorders [77].

Such studies are particularly important given the increase in shift work and long working hours over the past decade, as well as increased exposure to light and electronic devices at night due to work and social demands [82].

#### 4.1.2. Environmental Pollutants

The fetus is particularly susceptible to environmental contaminants as it develops during pregnancy and is, therefore, more susceptible to their effects. It is advised that pregnant women limit or prevent exposure to air pollution, especially during the early and late stages of pregnancy. Fast food, alcohol, and cigarette products, which are already known to be dangerous to human health, should be avoided by expectant mothers along with toxic natural items that include mycotoxins.

Pesticides, heavy metals, dioxin derivatives, and polychlorinated diphenyl compounds have been reported to have a significant effect on mutagenesis and teratogenesis, as have macroenvironmental pollutants, alcohol, drugs, and tobacco smoke [83,84]. The mechanism of action of these air pollutants is thought to be that pollutants such as heavy metals that reach the placenta change DNA methylation patterns, leading to changes in placental function and fetal reprogramming. In fact, reports on changes in placental DNA methylation associated with prenatal exposure to air pollution (including heavy metals) have shown that they are associated with changes in total DNA methylation and promoter DNA methylation of genes involved in biological processes such as energy metabolism, circadian rhythm, DNA repair, inflammation, cell differentiation, and organ development [85]. These issues are also of interest in relation to the DOHaD theory [79,86,87,88].

#### 4.1.3. DOHaD Theory

DOHaD is an acronym for Developmental Origin of Health and Disease.

Adverse environmental exposure during pregnancy can have a lifelong effect on offspring. For example, when the mother is subjected to nutritional restriction or prenatal stress, the fetus responds to adapt to the condition, which may improve the fetus’s adaptation to the postnatal situation in the short term, but in the long term, it is suggested that this may be the current cause and background of adult metabolic diseases such as cardiovascular disease, obesity, and metabolic syndrome (a condition in which visceral obesity is combined with high blood pressure, high blood sugar, and lipid metabolism disorders, making heart disease and strokes more likely to occur). This living environment is also thought to apply to the neonatal period after birth.

This is the concept that “future health and susceptibility to certain diseases are strongly influenced by the environment during the fetal period and early postnatal period.” This is the so-called DOHaD theory [79,86,87,88].

### 4.2. Newborns and Infants

#### 4.2.1. Premature Infants/Newborns

When premature infants (≥32 weeks of gestation) are raised on a regular light–dark schedule in the neonatal intensive care unit, they gain weight faster than infants raised under constant bright or dim light, effectively shortening their length of hospital stay [89,90]. Compared with infants raised under constant dim light, these infants cry less and are more active during the day [91]. A regular light–dark schedule accelerates the maturation of rest–activity, sleep/wake, and melatonin rhythms in premature infants [91,92,93].

Given the benefits of cycled lighting in the development of circadian rhythms, it should be implemented in both healthcare and home settings to promote optimum growth and development of the infant [94].

#### 4.2.2. Circadian Rhythms in Infancy

In the newborn period, waking every 3–4 h throughout the night indicates a proper ultradian rhythm [48,51,58]. However, parents need to feed their babies according to this rhythm, which fragments their sleep and makes daily life difficult. However, after two months of age, the baby is more likely to sleep for more than five or six hours, possibly even continuously through the night, which may make parenting easier overall (Figure 2A). On the other hand, frequent awakenings in babies after 2 months of age, especially after 3–4 months of age, indicate an interruption or persistence of ultradian rhythm and that the transition to the next important step, circadian rhythm formation, has not progressed properly. Infants who inherit various clocks in their organs during fetal development acquire a daily schedule of sleep and wake times while experiencing daily rhythms such as night and day and darkness and light and solidify the integration of their entire body’s biological clocks through this daily rhythm. It is believed that a baby’s circadian rhythm begins to become fixed between 6 months and 12 or 18 months of age [3,34,95,96], and, although there may be some margin of error, it is believed that the circadian clock is almost completely established in the suprachiasmatic nucleus by the age of 1.5~2, indicating that this is a critical period.

**Figure 2 clockssleep-07-00041-f002:**
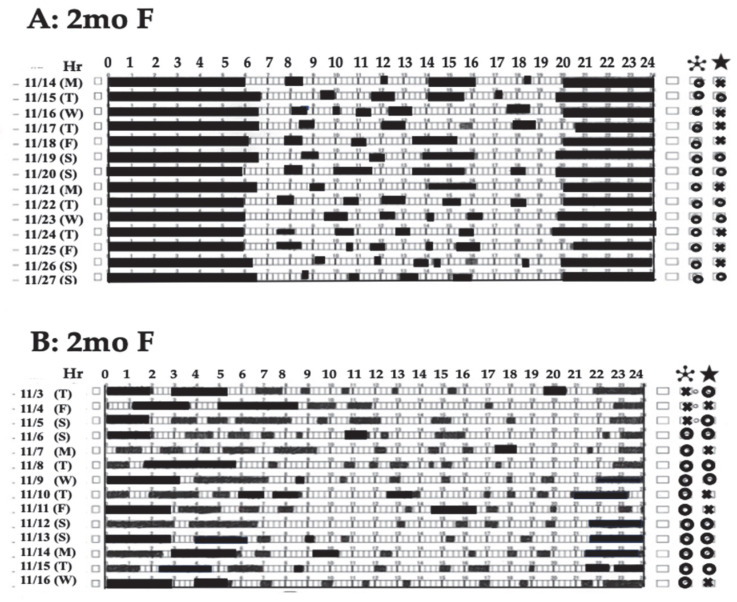
Example showing the development of nighttime sleep duration in 2-monthold-infants. In the top row (**A**), the infant is already sleeping continuously throughout the night. In the bottom row (**B**), the infant’s sleep is not sustained due to the influence of the habit of breastfeeding when falling asleep and every time the infant wakes up. 
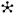
: Self-awakening, ★: Bow defecation.

As mentioned above, after the neonatal period, biological circadian rhythms begin to develop rapidly. During the first year of life, the sleep/wake rhythm continues to develop, coinciding with the increase in melatonin secretion at sunset [97]. In particular, the onset of nighttime sleep is linked to sunset for the first few months of life and then to the bedtime of the family. This suggests that the circadian rhythm is initially synchronized with light but is then synchronized with social and environmental factors.

After two months of age, as the circadian rhythms of hormones and thermoregulation in children begin to appear, sleep is concentrated at night, daytime wakefulness becomes longer, and nighttime sleep begins to last longer. It has been reported that nighttime sleep time is about 5–6 h at 2 months of age [98], about 8–9 h at 4 months of age [54], and about 8–12 h at 6–7 months of age [99]. Furthermore, it has been reported that children who can maintain sufficient sleep have a higher ability to self-soothe [98,99,100] and cry less at night. In order to foster the habit of self-soothing, parents need to refrain from immediately reaching out to help their baby when it cries and instead take the time to watch over it [101]. According to these reports, infants who can sleep through the night (8–12 h) between the ages of 4 and 6 months no longer need nighttime feedings [102].

In addition, the peripheral clocks work closely with the central clock to form a well-regulated network of circadian clocks, so it is recommended to eat breakfast, lunch, and dinner at approximately the same time for proper circadian rhythm synchronization [103,104].

#### 4.2.3. Breastfeeding/Nighttime Feeding

First, the secretion rhythm of the pineal hormone (melatonin), which is said to be the conductor of biological rhythms, begins at the end of the neonatal period [96]. Subsequently, the circadian rhythms of sleep and wakefulness and of thermoregulation also begin to appear at 2–3 months of age [55,104]. Since the formation of circadian rhythmicity in babies is surprisingly early, it is recommended that parents keep this in mind when raising their children.

Feeding (particularly breastfeeding) is often used as a method to encourage infants to fall asleep again after waking up. Breastfeeding is considered the optimal feeding method for newborns and their mothers.

In particular, the importance of breastfeeding during the neonatal period has been emphasized [105,106]. In addition, breast milk changes during lactation are perfectly adapted to the nutritional and immune needs of the infant, so it is recommended to continue breastfeeding until at least six months of age. However, the composition of breast milk changes throughout the day, and melatonin concentrations are higher in nighttime breast milk and are thought to be involved in the synchronization of circadian rhythms in children [107,108]. Therefore, it is not recommended to give breast milk produced during the day at night, and feeding habits adapted to the time are required. Circadian variations in some physiologically active compounds are thought to transmit chronobiological information from the mother to child and help the development of the biological clock [109,110]. On the other hand, it has been reported that breastfed infants tend to wake up more frequently at night [111,112], their mothers tend to get less sleep [113,114], and exclusive breastfeeding after birth can lead to vitamin D deficiency [115,116,117]. Therefore, careful consideration of the aforementioned facts is needed when relying solely on breastfeeding.

Frequent awakenings and long awakenings during nighttime sleep do not lead to the balanced mental and physical development of children, and the habit of breastfeeding every time a child wakes up at night may actually lead to the disruption of nighttime sleep continuity and disrupt the rhythm of the biological clock (Figure 2A,B). In other words, the frequent care of a crying or awake child during the night, especially care with food (breast milk, formula, water, etc.), has been reported to stimulate the peripheral biological clock and cause awakening reactions and sleep disorders and has attracted attention [118,119,120]. There is concern that the habit of breastfeeding a child every time he or she wakes up in the middle of the night may actually hinder the continuity of sleep throughout the night, leading to frequent and prolonged awakenings during nighttime sleep [119,121]. In infants aged 2 to 3 months, body movements may occur during REM sleep, which may be mistaken for waking up during the night. Since these body movements are often not a true awakening, it is thought that caregivers should not respond immediately but should help the child learn self-soothing to help them acquire the ability to fall asleep again naturally [102,122,123].

#### 4.2.4. Stopping Nighttime Breastfeeding

Authors recommend that nighttime feedings be discontinued within 6 months of age for the following reasons: First, because feeding-related biological clock activity is an important factor in maintaining the synchronization of circadian rhythms, it seems reasonable to avoid feeding during sleep hours regardless of age [14,15,16,43,44]. I believe that biological clock activity related to eating is a very important factor in synchronizing circadian rhythms and that it makes sense to avoid eating while sleeping regardless of age.

In fact, it is known that, at 4 months of age, the nighttime sleep time extends to 8–9 h, and breastfeeding during nighttime sleep is no longer necessary for physical or nutritional reasons [99,102,124]. This early cessation of nighttime breastfeeding has been reported to prevent future nighttime crying and sleep maintenance disorders, to maintain robust daily rhythms, to promote healthy physical and mental development [102], and to help reduce parental fatigue [102,121].

Frequent and long awakenings during nighttime sleep can distort a child’s brain function, so the more a parent touches their child during nighttime sleep, the more disrupted the child’s sleep will be, leading to sleep disorders [101,119,121]. This is because nighttime feeding activates the peripheral clocks and promotes wakefulness, which impairs the synchronization of the central clock function that recognizes nighttime as a time for sleep and rest. This is thought to lead to frequent awakenings at night and sleep continuity disorders (fragmentation). It is recommended to stop breastfeeding if nighttime crying becomes severe. However, it has been reported that early cessation of nighttime feedings reduces the occurrence of nighttime crying.

In our study [121], we found that if the habit of breastfeeding every time an infant wakes up continues throughout infancy, the infant is significantly more likely to suffer from sleep continuity disorders and develop ASD in the future.

#### 4.2.5. Development of Circadian Clocks and Naps

According to a report, infants take multiple naps in the morning, afternoon, and evening until around 7 months of age, and then one nap each in the morning and afternoon from 7 to 8 months onwards, and then one nap between 12:00 and 15:00 at 14 to 18 months of age [125]. In another report, a pattern of two naps per day is established by 9 to 12 months of age, and one nap in the afternoon is established by 15 to 24 months of age [126]. Furthermore, after the age of two, naps have been shown to delay the onset of nighttime sleep and reduce the quality and duration of sleep [127].

It has been reported that children who do not take naps appear as early as 2 years of age (2.5%); nearly 90% of children stop taking naps by the age of 5 years [127,128]; and, finally, most children do not take a nap by the age of 7 years [128].

Naps after the age of 2 years are due to delayed sleep onset, resulting in poorer quality of sleep and a shorter sleep duration, and if a sleep disorder is suspected in a preschool child, the nap situation should be considered [65]. In recent years, there have been reports of children over the age of three not taking naps, but the reasons for this are still unclear. However, there is data that suggests that if an NBSD of around 10 to 11 h is ensured, naps are not necessary for maintaining health. Naps are thought to be caused by the semi-daily rhythm of sleepiness from 12:00 to 15:00 and are considered to be a short break to refresh the brain tired from morning activities. It is said that the most gasoline is consumed when starting a car engine; similarly, the brain may also consume the most energy at the start of the day. The benefits of naps are generally known, but the reasons why they are taken remain unclear. Naps are thought to be a semi-circular sleep rhythm that is fundamentally different from the sleep that modern people take to make up for a lack of sleep, as they feel sleepy all day due to insufficient sleep at night.

#### 4.2.6. The Biological Clock Adapts to the School/Social Life Schedule

Considering that the clock mechanism is almost fixed by early infancy, it is easy to assume that the formation of a lifestyle rhythm that adapts to the long modern school/social life is an unavoidable and important matter. Specifically, considering school/social life, it is recommended to establish the habit of waking up on one’s own between 6 and 7 am and falling asleep between 8 and 9 pm to support this [102,129]. This lifestyle rhythm is the background and basis for the proper development of circadian clocks, so it is recommended to form a lifestyle rhythm that follows this basis.

Modern people live their school/social life, which requires a relatively strict and regular 24-h schedule. Globally, social life generally follows a typical morning rhythm, with activities starting around 8:00 am. However, the recent spread of late-night lifestyle habits among modern people, which has become globalized, has often caused a delayed life rhythm, including for infants and young children [129], creating a breeding ground for the so-called “social jet lag” [130,131,132,133]. It is obvious that such a shift in life schedule makes it difficult for humans to maintain proper activity and reduces their vitality. In addition to the physical and mental development of infants and young children, the following conditions are thought to be necessary for proper development of circadian clocks, which are involved in maintaining mental and physical health throughout one’s life [129,133]:

(1) Wake-up time: In the school/social life of the near future, students must be at school by 8:00 am. It is reported that children need about an hour from the time they wake up in the morning until they can start performing other things, which means they need to be up by 6 or 7 in the morning. If it takes a long time to get to school, they may have to wake up even earlier.

(2) Nighttime Basic Sleep Duration (NBSD): It is believed that there is a basic amount of nighttime sleep that children need from infancy through to toddlerhood. This NBSD varies slightly depending on race and region of residence, with Caucasian children reported to need 10–11 h (11 h is common) [34,134,135] and Japanese children just under 10 h [129]. It is more logical to determine whether a child is getting enough NBSD in order to determine whether they are suffering from a lack of sleep, which can be detrimental to them, and problems have been pointed out using the total amount of sleep per day as a criterion.

(3) Time of sleep onset: Based on the above conditions, the recommended time for sleep onset is between 7:00 and 9:00 pm, which is when children’s NBSD can be ensured before waking up in the morning.

(4) Duration of nighttime sleep: A condition for good-quality sleep is that nighttime sleep should be continuous, and frequent awakenings or awakenings lasting more than 30–60 min during nighttime sleep are known to be undesirable.

It is considered appropriate for a circadian clock that meets the above conditions to be mature by the time a child is around 1.5 to 2 years old.

## 5. Disruption of Biological Clocks

The circadian rhythm controls many important aspects of physiological functions, from the sleep–wake cycle to metabolism [3,4,5,6,7,8,9,10,11,12,13,14,15,16,17,18,19,20,21,22,23,24,25,26]. Any disruption of this rhythm impairs the organizing function that sustains life, and it is obvious that this will cause considerable problems throughout the body. In fact, disruption of circadian rhythms, including the sleep–wake rhythm, has been reported to cause systemic dysfunction at various stages of life [66,67]. The background to the disruption of circadian rhythms is thought to be prenatal factors, including genetic background, and environmental factors associated with the postnatal living environment. In this report, the following factors will be described as important points, with a focus on the fetal period, lactation, and early childhood.

### 5.1. Disruption of the Circadian Rhythms

#### 5.1.1. Jet Lag

The biological clock was originally thought to have a stable rhythm that was unlikely to go out of sync. It is said that the first time humans typically experience a disruption of the biological clock is when they experience jet lag while traveling abroad on a jet plane. It was not so long ago that humans were able to experience the workings of the biological clock. The biological clock was thought to be a relatively stable property of the human body that did not go out of sync easily, but, with the invention of airplanes and the evolution of jet planes, humans were able to travel to distant parts of the earth within a short time.

Jet lag occurs after crossing a time zone so rapidly that the circadian rhythm system cannot keep up, becomes out of sync with the day/night cycle of the destination [136,137,138]. The main symptoms of jet lag include insomnia, drowsiness, fatigue, headache, loss of appetite, and irritability, which affect almost the entire body. It is known that jet lag symptoms are milder when traveling westward. This is explained by the fact that the body clock is more adaptable to travel westward (where the daily cycle is longer) but less adaptable to eastward travel (where the daily cycle is shorter) [136]. If one is thrown into a living environment that suddenly reverses the rhythm of one’s biological clock in a short period of time, the biological clock cannot immediately synchronize, and the rhythm of one’s life with the outside world is not in sync, leading to a state in which the coordination between cells and organs is disrupted and the life support mechanisms cannot function (jet lag). In other words, the biological clock is a mechanism that supports life activities in accordance with the “night and day” of the place on Earth where humans live constantly. Furthermore, jet lag and shift work disorders have been found to induce circadian rhythm sleep/wake disruptions that arise from behavioral changes in sleep/wake schedules in relation to the external environment [137,139].

Jet lag and shift work sleep disorders are the results of dyssynchrony between the internal clock and the external light–dark cycle, brought on by rapid travel across time zones or by working a non-standard schedule. Symptoms can be minimized through optimizing the sleep environment, strategic avoidance of exposure to light, and drug and behavioral therapies [140].

#### 5.1.2. Social Jet Lag (SJL)

SJL means a mismatch between biological and social time. In other words, SJL is a misalignment between sleep and wake times on workdays and free days [129,130,131].

Our modern lifestyle and artificial nocturnal light delay our bedtime, make us wake up, and lead to a greater intraindividual variability in sleep timing. Depending on the constraints that social time places on us, our sleep timing may be in or out of phase with the internal circadian timing determined by the circadian clock. When a person’s social time is out of phase with their circadian time, they may be considered to suffer from circadian disruption or “social jet lag”. Symptoms include insomnia, drowsiness, fatigue, a heavy head, loss of appetite, and irritability. In addition, sleep and circadian rhythm disorders generally alter cognitive abilities (higher executive functions such as alertness, attention, memory, reaction inhibition, and decision-making). These can have a more serious impact on social life than the short-term problems associated with jet travel. Problems associated with the disruption of the biological clock in modern people have become serious problems across races and generations.

### 5.2. Chronodisruption

In this section, I discuss the rapid increase in the number of modern people who suffer from the jet lag state, which is similar to that experienced when traveling abroad but is more persistent, chronic, and malignant, despite living in the country or place where they were born and raised. The background to this may be, for example, the family’s lifestyle rhythm; however, it is also common that a child may have been living a “night owl” lifestyle since birth or may have a chronic lack of sleep and cannot adapt to the school/social rhythm of life due to their lifestyle and environment, such as homework loads, caffeine intake, early school start times, cramming school and other academic and sports activities, and increased TV and other screen time [141]. Recent studies have shown that long-term circadian rhythm disruption is associated with many pathological conditions, including premature death, obesity, impaired glucose tolerance, diabetes, mental illness, anxiety, depression, and cancer progression, while short-term disruption is associated with poor health, fatigue, and decreased concentration [142]. The existence of this condition has been recognized for some time and has been reported under various names; however, it is safe to assume that all of them refer to the same condition. The different names of this condition are as follows: (1) “biological (circadian) rhythm disturbance” [143,144], (2) “chronic fatigue syndrome” [145,146], (3) “chronodisruption” [147,148,149], (4) “social jet lag” [130], (5) “circadian misalignment” [150], (6) “circadian disruption” [151], and (7) “circadian syndrome” [152]. This condition, known as chronodisruption, is worth highlighting as it is closely related to our body clock.

As “chronodisruption” is thought to succinctly describe the pathology of this condition, this term is also used in this report. There are accumulating reports that this pathology, which can occur in any age group, is deeply related to the onset and worsening of other diseases, such as developmental disorders, school refusal, social withdrawal, glucose metabolism disorders (diabetes), depression, kidney disease, digestive diseases, cardiovascular diseases, dementia (Alzheimer’s disease), and cancer. Qian J et al. [43] provided a clear illustration of the pathology of time disruption in tissues throughout the body, which is a recommended reference. The importance of time disruption varies depending on the author; however, in severe cases, the author of this study personally believes that it would be more appropriate to call it “systemic chronodisruption,” considering that this pathology extends to the entire body.

### 5.3. Chronodisruption and DOHaD

Chronodisruption is a condition in which the biological clock mechanism does not mesh properly, resulting in systemic symptoms that disrupt the body’s life-sustaining functions [66,67,153]. In recent years, a number of reports, primarily from animal studies, have suggested that the lifestyle habits of pregnant mothers, such as sleep and meals, can affect the formation of the biological clock in their offspring and have lifelong effects on the offspring. In other words, it has been suggested that the qualities passed down by the mother during fetal development may be the background to health problems in various stages of life thereafter [154,155,156,157]. Human research has also begun to be reported, and the lifestyle habits of pregnant mothers have begun to attract attention [158,159]. For example, there are increasing reports that gestational chronodisorders caused by increasingly common exposure to irregular light may disrupt the circadian rhythm signals between the mother and the fetus and are likely to cause long-term health problems in the offspring. In fact, when examining the history of children with chronodisruption, it has been found that there is a high frequency of cases in which disruptions to daily rhythms are already present in the neonatal and infancy stages, including developmental disorders, school refusal, social withdrawal, and abnormal glucose metabolism (diabetes).

Recently, some studies have been published on the relationship between the mother’s living environment and the formation of the fetal circadian rhythm and how this may lead to various diseases in the future. These studies have reviewed recent information on the mother’s shift work, jet travel across time zones, irregular meals, and exposure to light at night during pregnancy that cause deviations and disruptions to the circadian clock, affecting fetal oxidative stress, the RAS (renin–angiotensin system), epigenetic regulation, and glucocorticoid programming, leading to various physical and mental health problems from infancy to adulthood (obesity, fatty liver, insulin resistance, kidney disease, hypertension, neurobehavioral disorders, reproductive dysfunction, cancer, and dementia) [66,67,153]. In other words, the research results indicate that problems in the formation of the circadian clock during fetal development may cause various health problems in adulthood, as stated in the DOHaD hypothesis. In addition to these health hazards, authors also report that developmental disorders are closely related to childhood problems such as “school refusal and social withdrawal due to circadian rhythm sleep disorders” [143,145,146]. In fact, many children with developmental disorders and school refusal after elementary school generally have a history of sleep disorders from the neonatal period. In other words, this supports the hypothesis that the misalignment or disruption of the biological clock is the basis of the DOHaD hypothesis and is likely to be an important research topic in the future. It has been reported that not only the mother’s but also the father’s lifestyle rhythms affect the formation of the offspring’s biological clock, and further research is needed [160].

Conversely, if it is possible to properly shape the biological clock to match the rhythm of modern society, it can be said that it is possible to maintain a relatively healthy life with minimal impairment of physical and mental functions throughout one’s life.

## 6. Preventing and Dealing with Deviations in Daily Rhythms

Finally, I will briefly discuss recommendations for daily life necessary to maintain appropriate daily rhythmicity and measures to take when deviations occur.

### 6.1. Daily Life of Pregnant Mothers

The following items are recommended:(1)Avoid staying up late (go to bed on the same day).(2)Ensure the amount of sleep and the time of day (ensure the amount of sleep required before waking up on weekdays).(3)Keep mealtimes consistent (avoid late-night snacks as much as possible).(4)Improve work style (avoid long hours and night shifts).(5)Avoid traveling abroad with large time differences.(6)Eliminate the effects of tobacco, alcohol, and other pollutants as much as possible.

If a deviation occurs in your daily rhythm, try to take measures such as regular melatonin and other preventive measures.

### 6.2. Life After Birth

The following items are recommended:(1)Establish a daily rhythm that allows you to ensure the basic nocturnal sleep duration (10–11 h: NBSD) according to the required wake-up time in the morning (usually before 7:00). For this purpose, a life rhythm in which the sleep onset time is set to before 7–9 p.m. in early infancy is recommended.(2)Nighttime feeding should be stopped in early infancy (2–6 months after birth) to foster sleep continuity in children.(3)Keep mealtimes consistent.(4)Correct the biological clock as soon as possible if it is out of sync.

### 6.3. Correcting the Child’s Life Rhythm

The following items are recommended:(1)Family therapy.

All family members should maintain a life rhythm in which they go to bed at an appropriate sleep onset time for 10–14 days.

(2)Pharmacological therapy.

If family therapy is ineffective, melatonin and other drugs may be used.

For details, please refer to other reports [161].

## 7. Conclusions

The circadian rhythm center, which begins to form during fetal development and is almost complete by early childhood, plays an important role in controlling life-sustaining functions such as sleep/wake, thermoregulation, and hormone secretion rhythms. Furthermore, it is known that circadian rhythms are deeply involved in the development of the mind and body and the maintenance of health throughout life. Therefore, it is known that disruption of the circadian rhythm can cause various dysfunctions throughout the body. This disruption of the body clock occurs at any stage of life, and this is related to predispositions associated with the influence of the mother during fetal development. These reports further support the DOHaD theory, suggesting that, along with nutritional problems, disruption of the body clock construction may be one of the core factors of DOHaD. Therefore, it is necessary to pay attention both personally and socially to the living environment of the mother during pregnancy, which is an important period for the formation of an appropriate circadian rhythm. After birth, the consolidation of circadian rhythms is greatly influenced by the qualities cultivated during fetal development as well as the daily living environment after birth, and the circadian rhythm center in the suprachiasmatic nucleus is almost completely formed during early childhood. Children’s daily rhythms during this important period are crucial for the development of their biological clocks, and parents are encouraged to raise their children with an interest in building a biological clock that will be suitable for maintaining their children’s future physical and mental health. Furthermore, regardless of the time, if you notice a deviation in your child’s daily rhythm, which has a significant impact on the development of their biological clock, correcting it immediately is an effective way to maintain appropriate daily rhythmicity and prevent the occurrence of chronodisruption, and it is advisable for those involved in childcare to be fully aware of the necessity of doing so.

## Figures and Tables

**Figure 1 clockssleep-07-00041-f001:**
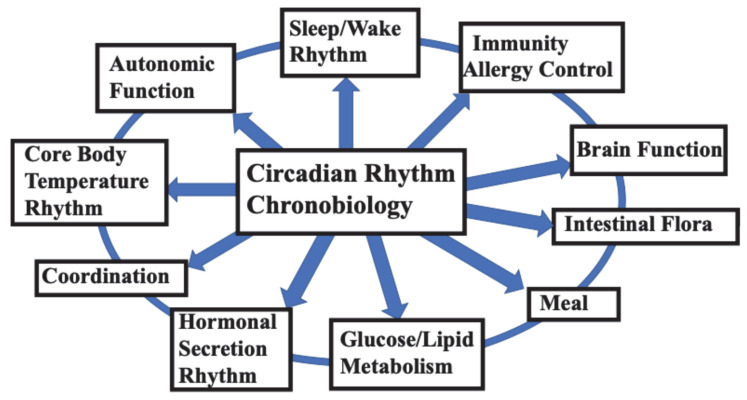
Circadian rhythm and life-sustaining functions. Circadian clocks are thought to be deeply involved in the life-sustaining functions that enable humans to live their daily lives [3,4,5,6,7,8,9,10,11,12,13,14,15,16,17,18,19,20,21,22,23,24,25,26].

## Data Availability

Data on infant nighttime sleep duration was obtained in collaboration with the ACC [129].

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
