# Peer review of "Appropriate Lifelong Circadian Rhythms Are Established During Infancy: A Narrative Review"

_2624-5175, 2025, doi:10.3390/clockssleep7030041_

Round 1
Reviewer 1 Report
Comments and Suggestions for Authors
Dear Authors,
Thank you for the opportunity to review your manuscript titled “Appropriate lifelong circadian rhythms are established during infancy.” The topic is highly relevant to the fields of chronobiology, developmental physiology, and early-life health programming. The manuscript touches on important aspects such as establishing fetal and neonatal circadian rhythms, the influence of maternal cues, and the implications of chronodisruption across the lifespan.
However, in its current form, the manuscript presents several challenges that significantly hinder its readability, clarity, and scientific contribution. I respectfully submit the following comments for your consideration:
- Structure and Clarity
- The manuscript is excessively long and repetitive, often restating similar concepts across multiple sections without a clear hierarchical organization.
- It blends physiological explanations, behavioral advice, and public health recommendations without consistently distinguishing between them.
- Several paragraphs suffer from imprecise or informal language (e.g., “tissues work together to tell the time”), which diminishes the academic tone.
Recommendation: A major structural reorganization is needed. Each section should have a clear thematic focus with logically ordered subheadings. Redundant content should be removed or consolidated.
- Objective and Scientific Contribution
- The manuscript currently lacks a clearly stated aim. It is unclear whether the main purpose is to:
- Review the developmental biology of circadian rhythms,
- Summarize clinical implications for infants and children,
- Or advocate behavioral recommendations for parents.
- As written, it reads more like a pedagogical essay or educational overview than a scientific review with a novel contribution.
Recommendation: The authors should explicitly state the review’s scope and objective in the abstract and introduction. Additionally, the discussion and conclusion should emphasize what is new, controversial, or integrative about this work.
- Use of References
- The manuscript relies heavily on a narrow subset of literature, particularly publications by a small group of authors.
- While these works are foundational, the limited diversity of references may give the impression of an insular perspective.
- Key recent findings from the broader circadian, pediatric sleep, and developmental programming literature are underrepresented.
Recommendation: I strongly encourage the authors to expand their references to include recent peer-reviewed work from diverse international groups and disciplines. A short list of suggested references is available upon request.
- Originality and Intellectual Attribution
A substantial portion of the manuscript’s structure and key arguments-such as:
- fetal SCN development and maternal entrainment,
- melatonin signaling during pregnancy,
- The ultradian-to-circadian rhythm transition in infants
- and the linkage with DOHaD closely mirror existing published reviews by specific research groups. While it is appropriate to build on foundational work, the manuscript must acknowledge its conceptual sources and demonstrate what is new, integrative, or expanded in this version.
Recommendation: Clarify how this review differs from previous ones. If the goal is to reframe known ideas for a new audience (e.g., clinical pediatrics), that aim should be transparently stated.
- Conclusion and Final Recommendation
Although the manuscript presents a rich topic with substantial educational value, its current form does not meet the standards of scientific clarity, critical synthesis, or originality required for publication in a peer-reviewed journal.
I therefore recommend major revisions. The authors are encouraged to:
- Streamline and restructure the text,
- Clearly articulate their contribution,
- Expand the reference base, and
- Appropriately acknowledge the intellectual origins of their framework.
If these substantial improvements are made, I would be happy to re-review a revised version.

No comment
Author Response
Dear reviewer
Thank you very much for your kind comments and suggestions for reviewing this document. I have revised the current version according to your comments. Below are your suggestions and my responses. I hope that it has been corrected appropriately. I am grateful for the many new papers you have introduced to me that I had not previously considered.
1. Recommendation: A major structural reorganization is needed. Each section should have a clear thematic focus with logically ordered subheadings. Redundant content should be removed or consolidated.
- response
I understand your point that it was long and repetitive. I have rewritten the paper throughout, removing as much repetition as possible and making it more concise.
- Recommendation:
Objective and Scientific Contribution
The manuscript currently lacks a clearly stated aim. It is unclear whether the main purpose is to:
- Review the developmental biology of circadian rhythms,
- Summarize clinical implications for infants and children,
- Or advocate behavioral recommendations for parents.
As written, it reads more like a pedagogical essay or educational overview than a scientific review with a novel contribution.
- response
The intent of this paper is 1) to summarize clinical implications for infants and young children, and 2) to propose behavioral recommendations for parents.
I have added the papers you have told me about and tried to rewrite it for this purpose. I have tried to add as many of the papers you have recommended to me as possible and rewrite it to meet these objectives.
3. Recommendation: The authors should explicitly state the review’s scope and objective in the abstract and introduction. Additionally, the discussion and conclusion should emphasize what is new, controversial, or integrative about this work.
Use of References
- The manuscript relies heavily on a narrow subset of literature, particularly publications by a small group of authors.
- While these works are foundational, the limited diversity of references may give the impression of an insular perspective.
- Key recent findings from the broader circadian, pediatric sleep, and developmental programming literature are underrepresented.
- response
The direction of this paper was explained as specifically as possible in the abstract and introduction. In the abstract and introduction, I explained the direction of this paper as specifically as possible. Regarding the points you pointed out, I added some of the reports you introduced, incorporated new perspectives from the paper, and rewrote it to more clearly introduce the DOHaD hypothesis. As a result, the direction of this paper is now focused on the health of the mother during pregnancy and the living conditions of the infant.
4. Recommendation: I strongly encourage the authors to expand their references to include recent peer-reviewed work from diverse international groups and disciplines. A short list of suggested references is available upon request.
Originality and Intellectual Attribution
A substantial portion of the manuscript’s structure and key arguments-such as:
- fetal SCN development and maternal entrainment,
- melatonin signaling during pregnancy,
- The ultradian-to-circadian rhythm transition in infants
- and the linkage with DOHaD closely mirror existing published reviews by specific research groups. While it is appropriate to build on foundational work, the manuscript must acknowledge its conceptual sources and demonstrate what is new, integrative, or expanded in this version.
- response
As I mentioned in part of my response to the previous comment. I won't go into detail, but I will try to address some of the points you mentioned.
- Conclusion and Final Recommendation
Although the manuscript presents a rich topic with substantial educational value, its current form does not meet the standards of scientific clarity, critical synthesis, or originality required for publication in a peer-reviewed journal.
I therefore recommend major revisions. The authors are encouraged to:
- Streamline and restructure the text,
- Clearly articulate their contribution,
- Expand the reference base, and
- Appropriately acknowledge the intellectual origins of their framework.
If these substantial improvements are made, I would be happy to re-review a revised version.
5.Respone
From the beginning, I have reviewed the structure and attempted to make major revisions. I have spent a lot of time reconstructing it, and I believe that I have mostly achieved my goal. I understand that it may be criticized as being biased, but this is the clinical perspective that I would like to present to you today.
Thank you for your kind comments and suggestions, I hope it has been corrected.
Reviewer 2 Report
Comments and Suggestions for Authors
Major Concerns:
- There are times when the author refers to “circadian rhythm” when they are actually talking about the circadian clock. It is important to note that “circadian clock” and “circadian rhythm” are two distinct concepts. The clock is the molecular feedback loop that establishes circadian rhythms. For example, the first sentence of the abstract references circadian rhythm when discussing the central clock in the SCN. This terminology needs to be corrected throughout the entire paper.
- Figure 2 is very blurry. I cannot read it at all. There also needs to be more explanation of the figure in the figure description so readers who are not familiar with looking at circadian actograms can understand what is in the figure. For example: what are the black boxes vs. white boxes.
- Lines 31-36, the author does not mention how the peripheral clocks also play a role in the rhythms of these physiological functions. The central clock does not regulate the rhythms of all these processes on its own. It is essential to note that the circadian clock mechanism is present in nearly every cell type and in tissues beyond the brain. They work together to create circadian rhythms. This is misleading to the readers.
- Lines 54-56: Light is not the only zeitgeber (entrainment cue). Food and exercise are also zeitgebers. This needs to be clarified for readers.
- Lines 76 – 79: Again, the central clock is not the only clock that contributes to physiological rhythms. There are plenty of studies showing that the peripheral clocks contribute, and this contribution is independent of the central clock.
- Line 388: DOHaD is not defined. They state that DOHaD is explained later, but it is never defined or explained. The author should also explain what they mean by “These issues are also of interest in relation to the DOHaD theory” on line 415 because they do not explain what DOHaD is. This seems like a big point of the paper, but without a description, the reader won’t know what this is.
- Line 513: I’m surprised the author did not mention “social jetlag” which is a well-studied circadian disruption similar to that of jetlag, which should also be discussed.
Minor Concerns:
- Is it necessary to abbreviate circadian rhythms to CR? This isn’t a typical abbreviation used in the field.
- Grammatical mistakes throughout
- Lines 27-29, does the author mean endogenous circadian rhythms when they state that CRs are observed even “at rest”? This should be clarified.
- Lines 41 – 42, the author should provide references to these resources.
- Lines 43-44: Author states that this review focuses on the relationship between circadian rhythms and ultradian rhythms, but this does not seem like the main point of the paper. UR is only mentioned in 2 areas.
- Lines 82-84: the author should provide references to these resources.
- Lines 98-99: ES cells also do not have a clock. Also, by “sex cells” does the author mean gametes? This should be clarified.
- Line 340 – is this supposed to be another subheading because it is not formatted like the others.
- Lines 343: human gene names should be capitalized and italicized.
- Line 511-512: Shouldn’t Zeitgebers be explained much earlier on in the paper when the authors discuss light/food/exercise cues.
- Abbreviations list: There are abbreviations used in the paper that are not on this list.
Significant grammar and terminology issues.
Author Response
Dear reviewer
Thank you very much for your kind comments and suggestions for reviewing this document. I have revised the current version according to your comments. Below are your suggestions and my responses. We hope that it has been corrected appropriately. I am grateful for the many new papers you have introduced to me that I had not previously considered.
1. Major Concerns:
There are times when the author refers to “circadian rhythm” when they are actually talking about the circadian clock. It is important to note that “circadian clock” and “circadian rhythm” are two distinct concepts. The clock is the molecular feedback loop that establishes circadian rhythms. For example, the first sentence of the abstract references circadian rhythm when discussing the central clock in the SCN. This terminology needs to be corrected throughout the entire paper.
- Response
I have revised my use of the terms circadian clock and circadian rhythm throughout the paper, and hopefully they are correct.
- Comment
Figure 2 is very blurry. I cannot read it at all. There also needs to be more explanation of the figure in the figure description so readers who are not familiar with looking at circadian actograms can understand what is in the figure. For example: what are the black boxes vs. white boxes.
2.response
Figure 2 has been reproduced.
3.Comment
Lines 31-36, the author does not mention how the peripheral clocks also play a role in the rhythms of these physiological functions. The central clock does not regulate the rhythms of all these processes on its own. It is essential to note that the circadian clock mechanism is present in nearly every cell type and in tissues beyond the brain. They work together to create circadian rhythms. This is misleading to the readers.
- response
I have rewritten it to avoid any misunderstanding.
- Comment
Lines 54-56: Light is not the only zeitgeber (entrainment cue). Food and exercise are also zeitgebers. This needs to be clarified for readers.
4.Response
I have added supplementary explanations, so I believe the description is more accurate. Since I have significantly restructured the paper, it is not possible to identify these parts of the description, so please judge the whole thing.
- Comment
Lines 76 – 79: Again, the central clock is not the only clock that contributes to physiological rhythms. There are plenty of studies showing that the peripheral clocks contribute, and this contribution is independent of the central clock.
- Response
I have added some papers and supplementary explanations, so I think the description is more accurate. Since I have significantly restructured the paper, it is not possible to identify these parts, so please judge it from the whole.
- Comment
Line 388: DOHaD is not defined. They state that DOHaD is explained later, but it is never defined or explained. The author should also explain what they mean by “These issues are also of interest in relation to the DOHaD theory” on line 415 because they do not explain what DOHaD is. This seems like a big point of the paper, but without a description, the reader won’t know what this is.
6. Response
I have added a section on DOHaD theory and provided an explanation.
7. Comment
Line 513: I’m surprised the author did not mention “social jetlag” which is a well-studied circadian disruption similar to that of jetlag, which should also be discussed.
7. Response
Although I have mentioned social jet lag in the form of a literature review, I have also provided a separate section to explain it.
Minor Concerns:
- Is it necessary to abbreviate circadian rhythms to CR? This isn’t a typical abbreviation used in the field.
Response:As per your suggestion, I have removed the abbreviation.
- Grammatical mistakes throughout
Response:I had asked an expert to proofread it, and now I have asked them to proofread it again.
- Lines 27-29, does the author mean endogenous circadian rhythms when they state that CRs are observed even “at rest”? This should be clarified.
Response: I deleted it because it was misleading.
- Lines 41 – 42, the author should provide references to these resources.
Response: I inserted the references.
- Lines 43-44: Author states that this review focuses on the relationship between circadian rhythms and ultradian rhythms, but this does not seem like the main point of the paper. UR is only mentioned in 2 areas.
Response: I have made it clear that this means that these are the only two that will be mentioned in this paper.
- Lines 82-84: the author should provide references to these resources.
Response: In the section Two Biological Clocks (Line 73-87), I explained the involvement of the two clock centers with references.
- Lines 98-99: ES cells also do not have a clock. Also, by “sex cells” does the author mean gametes? This should be clarified.
Response: I have decided that there is no need to go into detail about this issue and have therefore deleted it.
- Lines 343: human gene names should be capitalized and italicized.
Response: I deleted this item to shorten the length of the gene. Your comment was very helpful.
- Line 511-512: Shouldn’t Zeitgebers be explained much earlier on in the paper when the authors discuss light/food/exercise cues.
Response: I have moved the description as instructed.
- Abbreviations list: There are abbreviations used in the paper that are not on this list.
Response: I have corrected it according to your instructions.
Thank you for your kind comments and suggestions, I hope it has been corrected.
Reviewer 3 Report
Comments and Suggestions for Authors
Title: "Appropriate lifelong circadian rhythms are established during infancy"
This work aimed to describe factors associated with the development of the human circadian rhythms. The author wishes to describe “what is necessary for the proper development of biological clock, in order to promote a balanced mental and physical development of children and to maintain healthy state throughout their lives”.
The manuscript is interesting. The idea is potentially good. However, it sounds more like a book chapter instead of a review. The quoted literature is a lot, but not always up to date. Much of the news is not news but is only diligently ordered. The reader would expect to find some practical suggestions on how to promote the correct development of the circadian system, but there is nothing.
There is some naivety and some typos. For example, on page 2, the Author writes, “we can adapt to a comfortable daily life without any interference with our mental and physical functions”; however, no one yet knows how to quantify the cost of this adaptation, defined in some contexts as allostatic load. In the reference section, the number 47, the right name of the first Author is not Buruni but Bruni.
The quality of Figure 2 is very bad.
Author Response
Dear reviewer
Thank you very much for your kind comments and suggestions for reviewing this document. I have revised the current version according to your comments. Below are your suggestions and our responses. I hope that it has been corrected appropriately.
1. Comment
The quoted literature is a lot, but not always up to date. Much of the news is not news but is only diligently ordered. The reader would expect to find some practical suggestions on how to promote the correct development of the circadian system, but there is nothing.
- Response
An attempt has been made to review the literature and include the most up-to-date information, and some clinical approaches have also been mentioned.
- Comment
There is some naivety and some typos. For example, on page 2, the Author writes, “we can adapt to a comfortable daily life without any interference with our mental and physical functions”; however, no one yet knows how to quantify the cost of this adaptation, defined in some contexts as allostatic load. In the reference section, the number 47, the right name of the first Author is not Buruni but Bruni.
- Response
I think you're right. I thought this statement was a bit of an exaggeration so I deleted it.
Buruni has fixed it for Bruni.
Thank you for your kind comments and suggestions.
Reviewer 4 Report
Comments and Suggestions for Authors
This is a review on the infant formation of circadian rhythms, which can have future consequences for several diseases (both psychological and physiological) in adulthood. Developmental circadian biology is a growing and important topic, and it would benefit readers greatly if references are properly provided and facts and interpretations are clearly delineated.
There are instances where the authors omit references when introducing general ideas (e.g. lines 41-42, "please refer to the many well-known specialist books"; lines 83, "please refer to other relevant literature"). Please provide representative references for these general introductions.
The biological clock can refer to the maintenance of many different periodicities. Yet, the “CR biological clock” (lines 11, 27, and many more) is simply called the circadian clock. The SCN is therefore called the central circadian clock, not the “center of the biological clock” (line 86). It would be good to note that we have moved away from using the term “master-slave,” and the SCN is now less frequently referred to as the master clock. Similarly, "slave oscillator" (line 78) should be replaced with "peripheral oscillator", which has the same meaning.
There appears to be a misunderstanding about the “intracellular clocks” (lines 73–74 and 93). All circadian clocks are driven by molecular feedback loops, called transcription–translation feedback loops (TTFLs), which are intrinsically intracellular. Historically, there was a time when the clock was thought to be driven by membrane oscillations, but this was later proven to be incorrect. The molecular clock (again, intracellular) is now recognized as the core mechanism for all circadian clocks in eukaryotes. However, the authors claim that "some people consider the intracellular clock to be the third clock" (lines 74-75), without providing any reference. This needs to be corrected.
Especially in NICU biology, the entrainment of circadian rhythms is often a key concern. Therefore, it is important to be clear about the claims regarding entrainment cues. The authors state that “the circadian rhythm is initially synchronized with light, but then synchronized with social and environmental factors,” without providing any reference. This claim is not supported by the preceding citation [52] (the classic review by Rivkees). It appears to be based on the prior sentence: “the onset of night-time sleep is linked to sunset for the first few months of life, and then to the bedtime of the family”, again, presented without reference. However, this seems to reflect the author’s general impression rather than a scientific fact. These points need to be made more rigorous by clearly distinguishing between facts and opinions. Vague generalizations such as “social and environmental factors,” which cover all non-photic entrainment cues, should be avoided unless supported by evidence.
I am especially concerned about the claim that "... it is recommended to establish the habit of waking up on one's own between 6-7 am and falling asleep between 8-9 pm to support this [51, 58]." This recommendation does not accurately reflect the objectives of the original INSIGHT Responsive Parenting Intervention [51], which focused on promoting healthy sleep habits through responsive routines rather than prescribing fixed wake and sleep times.
For a rigorous scientific review, I believe casual observations and scientific evidence should be separated. The authors claim that “the brain may also consume the most energy at the start of the day” (line 335) because “the most gasoline is consumed when starting a car engine” (line 334). These two are clearly not equivalent. In fact, the brain does not consume the most energy upon waking. Instead, it maintains relatively high and consistent energy usage throughout the waking day, not by a sudden spike at wake onset.
There are also minor errors in the use of technical abbreviations. For human genes, the convention is to use capitalized italics. Therefore, in line 343, "Period (per), Clock (Clk), and cryptochrome (cry)" should be replaced with "Period (PER), Clock (CLK), and cryptochrome (CRY)".
The introduction to the suprachiasmatic nucleus is repeated in lines 508-509. There is a missing entry in the reference citation in line 97 ("[1, , 15-17]").
Comments on the Quality of English LanguageThere are several repetitions of the word definition, esp. "Suprachiasmatic nucleus (SCN)" (lines 33, 37, 509). "SCN in the hypothalamus" (line 86) and "SCN of the hypothalamus" (line 78) are other instances of repeated expressions. The writing can be streamlined to avoid repeating expressions and explanations unnecessarily.
Author Response
Dear reviewer
Thank you very much for your kind comments and suggestions for reviewing this document. I have revised the current version according to your comments. Below are your suggestions and my responses. I hope that it has been corrected appropriately.
- Comment
There are instances where the authors omit references when introducing general ideas (e.g. lines 41-42, "please refer to the many well-known specialist books"; lines 83, "please refer to other relevant literature"). Please provide representative references for these general introductions.
- Response
I've inserted some references.
- Comment
The biological clock can refer to the maintenance of many different periodicities. Yet, the “CR biological clock” (lines 11, 27, and many more) is simply called the circadian clock. The SCN is therefore called the central circadian clock, not the “center of the biological clock” (line 86). It would be good to note that we have moved away from using the term “master-slave,” and the SCN is now less frequently referred to as the master clock. Similarly, "slave oscillator" (line 78) should be replaced with "peripheral oscillator", which has the same meaning.
- Response
Thank you for pointing this out. Since it was a quote from a paper, I did not delete it, but I added the latest understanding to make it more accurate.
- Comment
There appears to be a misunderstanding about the “intracellular clocks” (lines 73–74 and 93). All circadian clocks are driven by molecular feedback loops, called transcription–translation feedback loops (TTFLs), which are intrinsically intracellular. Historically, there was a time when the clock was thought to be driven by membrane oscillations, but this was later proven to be incorrect. The molecular clock (again, intracellular) is now recognized as the core mechanism for all circadian clocks in eukaryotes. However, the authors claim that "some people consider the intracellular clock to be the third clock" (lines 74-75), without providing any reference. This needs to be corrected.
- Response
I understand your comment. I have added a new paper to explain the current state of the peripheral clock. I have deleted it because I am not claiming that it is a third clock, but simply repeating previous reports. I have deleted it because I am not claiming it to be a third watch, but simply repeating previous reports.
- Comment
Especially in NICU biology, the entrainment of circadian rhythms is often a key concern. Therefore, it is important to be clear about the claims regarding entrainment cues. The authors state that “the circadian rhythm is initially synchronized with light, but then synchronized with social and environmental factors,” without providing any reference. This claim is not supported by the preceding citation [52] (the classic review by Rivkees). It appears to be based on the prior sentence: “the onset of night-time sleep is linked to sunset for the first few months of life, and then to the bedtime of the family”, again, presented without reference. However, this seems to reflect the author’s general impression rather than a scientific fact. These points need to be made more rigorous by clearly distinguishing between facts and opinions. Vague generalizations such as “social and environmental factors,” which cover all non-photic entrainment cues, should be avoided unless supported by evidence.
4.Response                                             I understand your point. I wanted to avoid ambiguity so I deleted it.
- Comment
I am especially concerned about the claim that "... it is recommended to establish the habit of waking up on one's own between 6-7 am and falling asleep between 8-9 pm to support this [51, 58]." This recommendation does not accurately reflect the objectives of the original INSIGHT Responsive Parenting Intervention [51], which focused on promoting healthy sleep habits through responsive routines rather than prescribing fixed wake and sleep times.
5.Response
This problem is understood slightly differently. My argument is based on the fact that as long as humans continue to live in a morning-oriented society, they cannot ignore this lifestyle schedule, and I have seen countless cases of children who deviate from this rule having difficulties in school and social life, and on clinical evidence that this causes an imbalance in the mental and physical functions of infants and young children. Therefore, I have no intention of changing this. I would like you to understand that this is exactly what I am saying.
- Comment
For a rigorous scientific review, I believe casual observations and scientific evidence should be separated. The authors claim that “the brain may also consume the most energy at the start of the day” (line 335) because “the most gasoline is consumed when starting a car engine” (line 334). These two are clearly not equivalent. In fact, the brain does not consume the most energy upon waking. Instead, it maintains relatively high and consistent energy usage throughout the waking day, not by a sudden spike at wake onset.
6.Response
Thank you for pointing this out. This sentence contained my own speculation and appears to have been incorrect. I will delete it.
7.Comment
There are also minor errors in the use of technical abbreviations. For human genes, the convention is to use capitalized italics. Therefore, in line 343, "Period (per), Clock (Clk), and cryptochrome (cry)" should be replaced with "Period (PER), Clock (CLK), and cryptochrome (CRY)".
- Response
I received the same coment from other reviewers. Some of them said it was too long, so I thought it was not an essential part and deleted it.
- Comment
The introduction to the suprachiasmatic nucleus is repeated in lines 508-509. There is a missing entry in the reference citation in line 97 ("[1, , 15-17]").
- Response
It has been corrected.
- Comment
There are several repetitions of the word definition, esp. "Suprachiasmatic nucleus (SCN)" (lines 33, 37, 509). "SCN in the hypothalamus" (line 86) and "SCN of the hypothalamus" (line 78) are other instances of repeated expressions. The writing can be streamlined to avoid repeating expressions and explanations unnecessarily.
- Response
Thank you for pointing that out. I hope I can fix it.
Thank you for your kind comments and suggestions, I hope it has been corrected.
Round 2
Reviewer 1 Report
Comments and Suggestions for Authors
The authors have significantly revised the manuscript, improving its structure and clarity.
Author Response
Dear reviewer
Thank you very much for your kind comments and suggestions for reviewing this document. I have revised the current version according to your comments. Below are your suggestions and my responses. I hope that it has been corrected appropriately. I appreciate your precise feedback. Thank you.
The yellow shading in the returned manuscript indicates the parts that have been rewritten based on the reviewer's suggestions.
The following statements have been deleted based on the reviewer's advice.
Line 58. “This fact is now widely accepted” is deleted.
Line 70. “and control” is deleted.
Line 78-79. “of the hypothalamus in the brain” is deleted.
Line129-130. “It is gradually becoming clear that the formation of the circadian clock is closely related to brain development.” Is deleted.
Line 136. “Biological” is deleted.
Line 169. Reference [77]: Replaced with the literature you pointed out.
Line 223-225 “It is the idea that "future health and susceptibility to certain diseases are strongly influenced by the environment during the fetal period and early postnatal period.” Is deleted.
Line 266. “biological” is deleted.
Line 267. “T” is deleted.
Line 329. “to consider that it is reasonable” is deleted.
Line 426. “as shown in Figure 1” is deleted.
Line 441. “when the biological rhythm, which occurs” is deleted.
Changed reference number
[48]→[50]
[49]→[48]
[50]→[49]
Reference [77]: Replaced with the literature you pointed out.
[125] is deleted
Below, 126 is reduced by 125, 127 by 126, and so on. The green shading indicates the new number that have been rewritten.
Thanks again, and I hope this manuscript has been appropriately improved.
